# The genome of an apodid holothuroid (*Chiridota heheva*) provides insights into its adaptation to a deep-sea reducing environment

Long Zhang[1,7], Jian He[1,7], Peipei Tan[1], Zhen Gong[2], Shiyu Qian[3], Yuanyuan Miao[1], Han-Yu Zhang[4], Guangxian Tu[1], Qi Chen[1], Qiqi Zhong[1], Guanzhu Han[2], Jianguo He [1,5,6✉] & Muhua Wang [1,6✉]

Cold seeps and hydrothermal vents are deep-sea reducing environments that are characterized by lacking oxygen and photosynthesis-derived nutrients. Most animals acquire nutrition in cold seeps or hydrothermal vents by maintaining epi- or endosymbiotic relationship with chemoautotrophic microorganisms. Although several seep- and vent-dwelling animals hosting symbiotic microbes have been well-studied, the genomic basis of adaptation to deep-sea reducing environment in nonsymbiotic animals is still lacking. Here, we report a high-quality genome of *Chiridota heheva* Pawson & Vance, 2004, which thrives by extracting organic components from sediment detritus and suspended material, as a reference for nonsymbiotic animal's adaptation to deep-sea reducing environments. The expansion of the aerolysin-like protein family in *C. heheva* compared with other echinoderms might be involved in the disintegration of microbes during digestion. Moreover, several hypoxia-related genes (Pyruvate Kinase M2, *PKM2*; Phospholysine Phosphohistidine Inorganic Pyrophosphate Phosphatase, *LHPP*; Poly(A)-specific Ribonuclease Subunit PAN2, *PAN2*; and Ribosomal RNA Processing 9, *RRP9*) were subject to positive selection in the genome of *C. heheva*, which contributes to their adaptation to hypoxic environments.

[1] State Key Laboratory for Biocontrol, Southern Marine Science and Engineering Guangdong Laboratory (Zhuhai), School of Marine Sciences, Sun Yat-sen University, Zhuhai 519000, China. [2] College of Life Sciences, Nanjing Normal University, Nanjing 210023, China. [3] School of Medicine, Jinan University, Guangzhou 510632, China. [4] Hainan Key Laboratory of Marine Georesource and Prospecting, Institute of Deep-sea Science and Engineering, Chinese Academy of Sciences, Sanya 572000, China. [5] Guangdong Province Key Laboratory for Aquatic Economic Animals, School of Life Sciences, Sun Yat-sen University, Guangzhou 510275, China. [6] Maoming Branch, Guangdong Laboratory for Lingnan Modern Agricultural Science and Technology, Maoming 525435, China. [7] These authors contributed equally: Long Zhang, Jian He. ✉email: lsshjg@mail.sysu.edu.cn; wangmuh@mail.sysu.edu.cn

Echinodermata is a phylum of marine animals comprising 5 extant classes, including Holothuroidea (feather star, subphylum Pelmatozoa), Asteriodea and Ophiuroidea (starfish and brittle star, subphylum Asterozoa), and Echinoidea and Holothuroidea (sea urchin and sea cucumber, subphylum Echinozoa)[1]. Adult echinoderms are characterized by having a body showing pentameral symmetry, a water vascular system with external tube feet (podia), and an endoskeleton consisting of calcareous ossicles[2]. Echinoderms exhibit a high divergence in morphology, from the star-like architecture in Asteroidea to the worm-like architecture in Holothuroidea[3,4].

Compared with other echinoderms, holothurians have a unique body architecture and evolutionary history. The worm-like body of the holothurian preserves the pentameral symmetry structurally along the oral–aboral axis[5]. In addition, holothurians have a soft and stretchable body, in which the ossicles are greatly reduced in size[2]. The order Apodida is a group of holothurians that are found in both shallow-water and deep-sea environments[6]. Phylogenetic analyses showed that Apodida is sister to other orders of Holothuroidea[7,8]. Apodid holothurians lack tube feet and complex respiratory trees, making them morphologically distinct from other holothurians[2]. In contrast to other classes of Echinodermata, which experienced a severe evolutionary bottleneck during the Permian-Triassic mass extinction interval, Holothuroidea did not experience family-level extinction through the interval. The deposit-feeding lifestyle of holothurians conferred a selective advantage during the primary productivity collapse of the Permian–Triassic mass-extinction[9]. As the genomes of only two shallow-water holothurians (*Apostichopus japonicus* and *Parastichopus parvimensis*) have been assembled and analyzed[10–12], it is critical to study the genomes of more holothurians to dissect their special morphological characteristics and evolutionary history.

Cold seeps are areas where methane, hydrogen sulfide, and other hydrocarbons seep or emanate as gas from deep geologic sources[13]. Hydrocarbon-fluid seepage from cold seeps is completely devoid of $O_2$ and comprises high levels of sulfides. After reacting with sulfides contained in the fluid, any free $O_2$ is removed from the deep-sea water. Thus, cold seeps are characterized by high hydrostatic pressure, low temperature, lack of oxygen, and photosynthesis-derived nutrients, and high concentrations of reducing chemicals[14]. Chemosynthetic microbes oxidize the reduced chemicals contained in the hydrocarbon fluids to produce energy and fix carbon into organic matter, which supports a large amount of invertebrates, including

tubeworms, mussels, clams, and gastropods[15]. Most of these macrobenthos depend on the epi- or endosymbiotic relationships with chemoautotrophic microorganisms for nutrition[14,16,17]. Recent genomic analyses have revealed the genetic basis of adaptation in several seep- and vent-dwelling macrobenthos hosting symbiotic bacteria[18–21]. However, the genomic basis of nutrient acquisition and hypoxic adaptation in cold seep-dwelling nonsymbiotic animals is still lacking with only one reported genome[22].

Echinoderms are a rare component of deep-sea chemosynthetic ecosystems[23]. *Chiridota heheva* Pawson & Vance, 2004 (Apodida: Chiridotidae) is one of the few echinoderms that occupies all three types of chemosynthetic ecosystems (hydrothermal vent, cold seep, and organic fall)[24]. This suggests that the species is well adapted to deep-sea reducing environments. Unlike most cold seep- and hydrothermal vent-dwelling species, *C. heheva* does not host chemosynthetic bacteria[6]. It derives nutrients from a variety of sources, extracting organic components from sediment detritus, suspended material, and wood fragments when available[6,25]. The cosmopolitan distribution and special lifestyle of *C. heheva* make it an ideal model to study adaptation to deep-sea reducing environments in nonsymbiotic animals.

Here, we assembled and annotated a high-quality genome of *C. heheva* collected from the Haima cold seep in the South China Sea. Evolutionary analysis revealed that the ancestor of *C. heheva* diverged from the ancestors of two shallow-water holothurians (*A. japonicus* and *P. parvimensis*) approximately 429 Ma ago. Additionally, demographic analysis suggested that *C. heheva* might have colonized the current habitat in the early Miocene. Comparative genomic analysis showed that the aerolysin-like protein (ALP) family was significantly expanded in *C. heheva* compared with other echinoderms. The expansion of the ALP family might be involved in the disintegration of microbes during digestion, which in turn facilitated its adaptation to cold seep environments. Moreover, several hypoxia-related genes were subject to positive selection in the genome of *C. heheva*, which contributes to their adaptation to hypoxic environments.

## Results and discussion

**Characterization and genome assembly of *C. heheva*.** The sequenced sample was collected at a depth of 1385 meters using manned submersible *Shenhaiyongshi* from the Haima cold seep in the South China Sea (16° 73.228′ N, 110° 46.143′ E) (Fig. 1). We sequenced the sample's genome on the Nanopore and Illumina

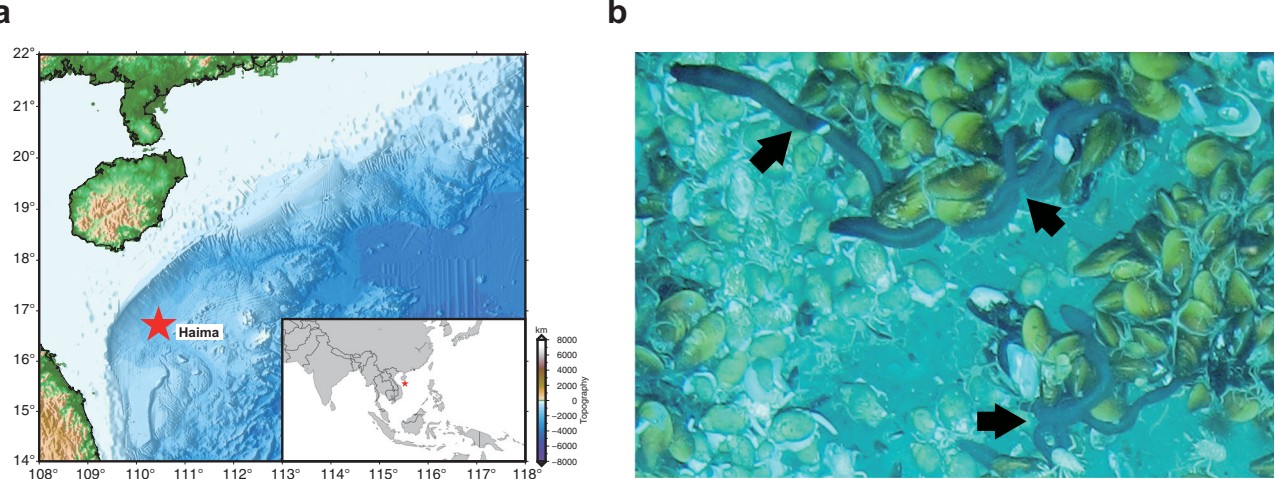

**Fig. 1 Collection of *C. heheva*. a** Map showing the sampling site at the Haima cold seep of northern South China Sea (16° 73.228′ N, 110° 46.143′ E). **b** *C. heheva* at the sampling site (depth: 1385 m), where they cohabit with deep-sea mussels. *C. heheva* individuals are indicated by black arrows. Photo by Dr. Jian He.

**Table 1 Genome assembly statistics of deep-sea holothurian (*C. heheva*) and shallow-water holothurian (*A. japonicus*).**

|  | *C. heheva* | *A. japonicus*[11] | *A. japonicus*[10] |
|---|---|---|---|
| Estimated genome size (Gb) | ~ 1.23 | ~ 1.0 | ~ 1.0 |
| Assembled genome size (bp) | 1,106,937,276 | 952,279,490 | 804,993,085 |
| Number of contigs | 4609 | 21,303 | 7058 |
| Contig N50 (bp) | 1,221,604 | 45,411 | 190,269 |
| Scaffold N50 (bp) | – | 195,518 | 486,650 |

sequencing platforms. In total, 42.43 Gb of Nanopore reads and 49.19 Gb of Illumina reads were obtained (Supplementary Tables 1 and 2). Species identity of the sequenced individual was first determined according to its morphological characteristics. In addition, we assembled the mitochondrial genome of the individual using Illumina reads. The sequence identity between the published *C. heheva* mitochondrial genome and our assembled genome was 99.74%, which confirmed the species identity of the sequenced individual[26]. Based on the *k*-mer distribution of Illumina reads, the size of the *C. heheva* genome was estimated to be 1.23 Gb with a high heterozygosity of 2% (Supplementary Fig. 1 and Supplementary Table 3). The *C. heheva* genome was assembled into 4609 contigs, with a total size of 1.107 Gb and contig N50 of 1.22 Mb (Table 1). We determined the completeness of the assembled genome by running benchmarking universal single-copy orthologs (BUSCO) and sequencing quality assessment tool (SQUAT) software. BUSCO analysis with metazoan (obd10) gene set showed that the assembled *C. heheva* genome contained 89.6% complete single-copy orthologs (Supplementary Table 4). Additionally, 91.1% of Illumina reads could be aligned to the assembled genome with high confidence in SQUAT assessment (Supplementary Table 5). These results indicate the high integrity of our assembled genome.

**Genome annotation.** Repetitive elements represented 624.38 Mb in the *C. heheva* genome assembly (Supplementary Table 6). Long interspersed nuclear elements (LINEs) were the largest class of annotated transposable elements (TEs), making up 9.72% of the genome. DNA transposons, which were the second largest class of TEs, represented 33.59 Mb (3.03%) of the genome. Additionally, the *C. heheva* genome comprised a large proportion (38.39%) of unclassified interspersed repeats. Comparative genomic analysis among *C. heheva* and other echinoderms revealed that the *C. heheva* genome consisted of the largest number of TEs (Fig. 2a, b; Supplementary Table 7). Repetitive elements constituted 56.40% of the *C. heheva* genome, and they accounted for 26.68% and 25.02% of the genomes of *A. japonicus* and *P. parvimensis*, respectively. The differences in the repeat content were close to the size differences between the genomes of *C. heheva* and the other two holothurians. This suggests that repeats contributed to the size differences among the genomes of these three holothurians. Notably, the proportion of LINEs in the *C. heheva* genome was substantially higher than that in the genomes of other echinoderms (Fig. 2b). Kimura distance-based copy-divergence analysis identified a recent expansion of LINEs in the *C. heheva* genome (Fig. 2c). Protein-coding genes were identified in the genome of *C. heheva* through a combination of ab initio and homology-based protein-prediction approaches. In total, we derived 36,527 gene models in the *C. heheva* genome. The structure of predicted genes in *C. heheva* is slightly different to that of other previously sequenced echinoderm genomes. With

longer exons and introns, genes in *C. heheva* are longer than the ones in *A. japonicus* (Supplementary Table 8).

**Phylogenomic analysis and demographic inference.** With more than 1400 extant species, Holothuroidea is one of the largest classes in the phylum Echinodermata[1]. In addition, holothurians are well adapted to diverse marine environments, with habitats ranging from shallow intertidal areas to hadal trenches[27,28]. However, due to the lack of body fossils, evolutionary study of Holothuroidea is more difficult than other classes of Echinodermata. To investigate the evolutionary history of *C. heheva*, a maximum-likelihood (ML) phylogenetic tree was reconstructed using single-copy orthologs of *C. heheva* and 16 other deuterostomes (Supplementary Fig. 2). *Chiridota heheva* appeared sister to two other holothurians. In addition, divergence times were determined among 7 echinoderms that had whole-genome sequences (Fig. 3a). The divergence time of *A. japonica* and other echinoderms was estimated to be approximately 539 million years (Ma), which is generally consistent with the fossil records[29,30]. The ancestor of *Chiridota heheva* diverged from the ancestors of two shallow-water holothurians (*A. japonicus* and *P. parvimensis*) approximately 429 Ma ago. As Apodida is the basal taxon in Holothuroidea, these results support the view that holothurians had evolved by the Early Ordovician[31].

To better investigate the evolution of holothurians, we inferred the histories of ancestral-population sizes of *C. heheva* and *A. japonicus* using the pairwise sequential Markovian coalescent (PSMC) method (Fig. 3b). *Chiridota heheva* experienced a decline in population size approximately 21 Ma ago. Ocean temperature increased slowly between the late Oligocene and early Miocene (21–27 Ma ago) after long-term cooling from the end of the Eocene[32,33]. Furthermore, species diversity within Echinodermata started to increase in the early Miocene[34,35]. These results indicate that *C. heheva* might have colonized the current habitat in the early Miocene. A decline in ancestral-population size in *A. japonicus* started in the late Miocene (approximately 8 Ma ago). *Chiridota heheva* also experienced a moderate decline in population size in the early Pliocene. Additionally, the oceans experienced a decrease in temperature during the late Miocene (7–5.4 Ma ago)[36]. These results suggest that global cooling and environmental changes in the late Miocene were an important driver of demographic changes in both shallow-water and deep-sea holothurians.

**Hox/ParaHox gene clusters.** Apodida do not have tube feet or complex respiratory trees, which are commonly found in other holothurians[37]. It has been demonstrated that *Hox* genes play a critical role in embryonic development[38]. In addition, previous studies proposed that the presence/absence and expression pattern of *Hox* genes might contribute to morphological patterning and embryonic development in echinoderms[10,11]. Therefore, to determine whether *Hox* genes contribute to morphological divergence in Holothuroidea, we identified *Hox* gene clusters and their evolutionary sister complex, the *ParaHox* gene cluster, in the genomes of *C. heheva* and 6 other echinoderms (Supplementary Fig. 3). A *Hox* cluster and a *ParaHox* cluster could be identified in the genomes of all 7 species. The gene composition and arrangement of both *Hox* and *ParaHox* clusters were highly consistent between the genomes of *C. heheva* and *A. japonicus*, suggesting that *Hox/ParaHox* genes do not control the development of tube feet and respiratory trees in Apodida. *Hox4* and *Hox6* were missing in the genomes of both *C. heheva* and *A. japonicus*, which is inconsistent with the view that the loss of *Hox4* or *Hox6* in echinoderms is a lineage-specific event[5].

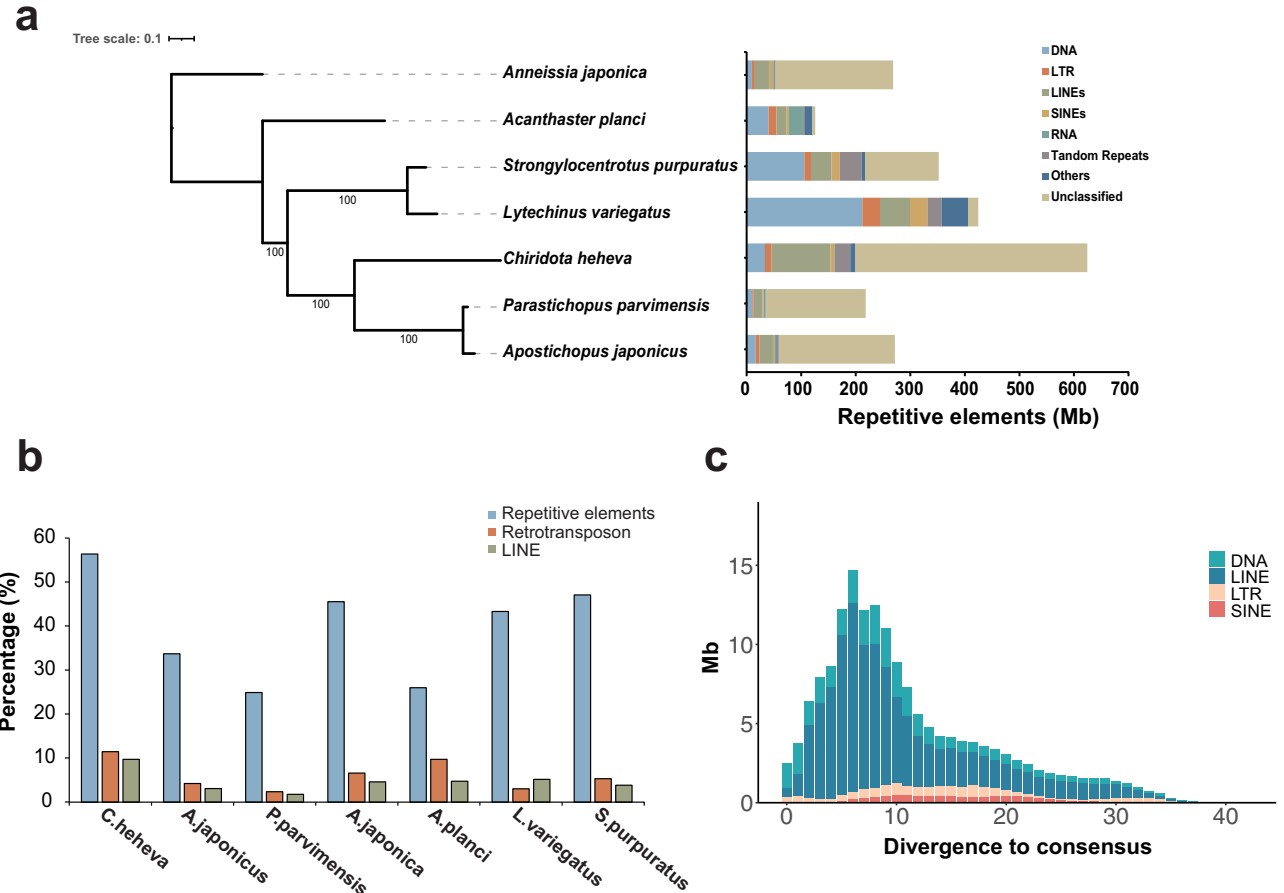

**Fig. 2 Landscape of transposable elements in echinoderm genomes. a** Comparison of the occurrence and composition of repetitive elements in the genomes of 7 echinoderms. **b** Comparison of the proportion of repetitive elements, retrotransposon, and long interspersed nuclear elements (LINEs) in the genomes of 7 echinoderms. The proportions of repetitive elements and LINEs are higher in the genome of *C. heheva* than that in other echinoderms. **c** Transposable element-accumulation profile in *C. heheva* genome. A recent burst of LINEs was observed in *C. heheva*.

**NLR repertoire in *C. heheva*.** NACHT and leukine-rich, repeat-containing proteins (NLRs) are important components of pathogen-recognition receptors (PRRs) involved in animal innate immune systems, which can perceive pathogen-associated molecular patterns (PAMPs) of viruses and bacteria[39]. The bona fide NLRs contain a NACHT (NAIP, CIITA, HET-E, and TP1) domain, which belongs to the signal transduction ATPases with numerous domain (STAND) superfamilies, and a series of C-terminal leukine-rich repeats (LRRs)[40,41]. The Pfam hidden Markov model (HMM) search combined with phylogenetic analysis approach identified only 53 NLRs in *C. heheva* (Supplementary Table 9), compared with a largely expanded set of 203 NLRs in purple-sea urchin (*Strongylocentrotus purpuratus*)[42]. *Chiridota heheva* contained 24 NLRs with one or more N-terminal death/DED domain, 22 NACHT-only NLRs, 6 NLRs with other domains, including the immunoglobulin V-set domain, which was not identified in sea-urchin NLRs, and only one NLR with LRRs (Supplementary Table 9). Taken together, these results indicate that the *C. heheva* NLR repertoire shows different abundances and structural complexities than the sea urchin.

We performed phylogenetic analysis of *C. heheva* NLRs and other representative NLRs of metazoans, including humans, *Amphimedon queenslandica*, *S. purpuratus*, *Acropora digitifera*, *Nematostella vectensis*, *Pinctada fucata*, *Capitella teleta*, mollusks, and arthropods[43]. We found that the majority of *C. heheva* NLRs form a monophyletic lineage with sea-urchin NLRs (Fig. 4), supporting the lineage-specific evolution of NLRs

in Echinodermata[44]. Given that human IPAF (ice protease-activating factor) and NAIP (neuronal apoptosis-inhibitory protein) proteins were reported to have originated before the evolution of vertebrates[44], one *C. heheva* NLR clustering with these two proteins indicates that this NLR may have an ancient independent origin (Fig. 4).

**Gene-family evolution.** We performed gene-family analysis based on the phylogenetic tree of 7 echinoderms (Fig. 3a). Compared with other echinoderms, 66 gene families were expanded, and 25 gene families were contracted in *C. heheva* (*P* < 0.05) (Supplementary Data 1 and Supplementary Table 10). Several significantly expanded gene families are involved in the processes of cell cycle progression, protein folding, and ribosome assembly. As high hydrostatic pressure causes cell cycle delay and affects protein folding[45,46], expansion of these families may have contributed to the adaptation of *C. heheva* to cold seep environments.

Aerolysins, which are pore-forming toxins (PFTs), were first characterized as virulence factors in the pathogenic bacterium *Aeromonas hydrophyla*[47,48]. As typical pore-forming proteins, aerolysin and related proteins are found in a large variety of species and possess diverse functions[49]. ALPs in eukaryotes originated from recurrent horizontal gene transfer (HGT)[50]. ALPs of the same origin have similar functions, while the ones of different origins possess diverse functions[50]. The ALPs were

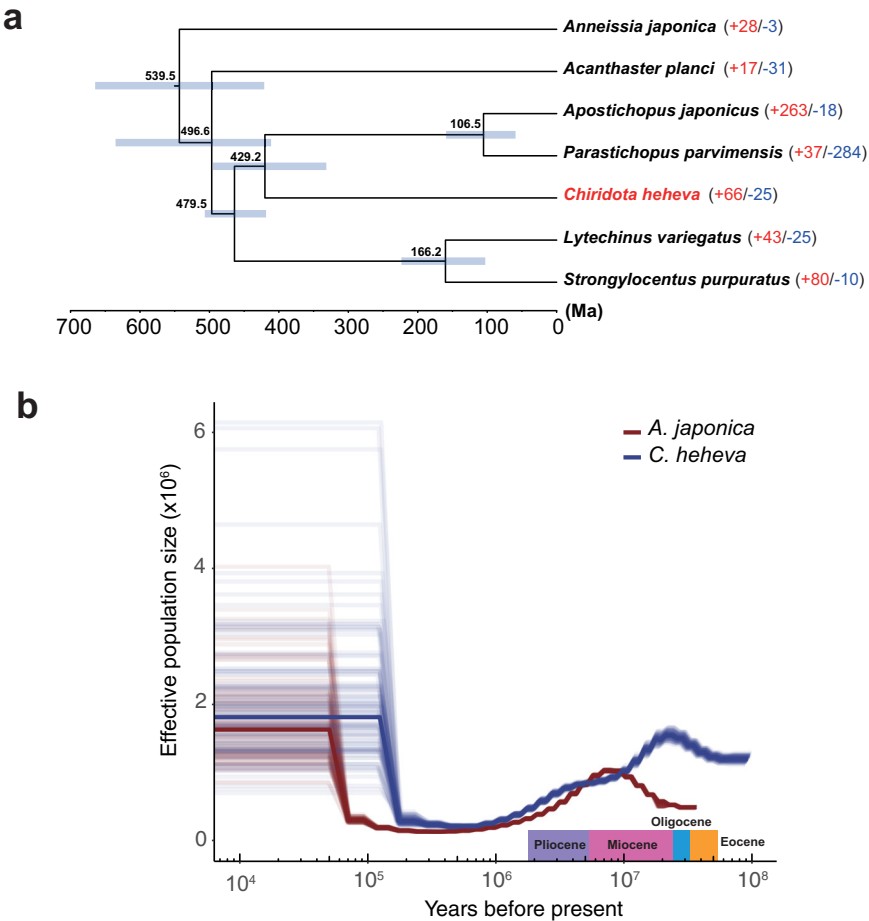

**Fig. 3 Evolutionary history of *C. heheva*. a** A species tree of 7 echinoderm species. In total, 988 single-copy orthologs were used to reconstruct the phylogenetic tree. The divergence time between species pairs was listed above each node, and 95% confidence interval of the estimated divergence time was denoted as blue bar. The numbers of protein families that were significantly expanded (red) and contracted (blue) (*P* < 0.05) in each species are denoted beside the species names. **b** Demographic history of *C. heheva* (blue) and *A. japonicus* (red). The changes of ancestral-population size of *C. heheva* and *A. japonicus* were inferred using the PSMC method. Time in history was estimated by assuming a generation time of 3 years and a mutation rate of $1.0 \times 10^{-8}$.

significantly expanded in the genome of *C. heheva* (7 copies) compared with other echinoderms (0 or 1 copy) (*P* < 0.05) (Supplementary Data 1). To investigate the possible origin and function of *C. heheva* ALPs, we reconstructed the phylogenetic tree of ALPs in echinoderms and diverse species. *Chiridota heheva* ALPs did not cluster with ALPs from other echinoderms. Additionally, these two groups of ALPs were clustered with aerolysins from distinct groups of bacteria (Fig. 5). This suggests that ALPs from *C. heheva* and other echinoderms have different origins. *Chiridota heheva* ALPs form a clade with ALPs from stony corals (*Stylophora pistillata*, *Pocillopora damicornis*, and *Orbicella faveolata*) and sea anemones (*Nematostella vectensis* and *Ecaiptasia diaphana*). This indicates that ALPs from *C. heheva*, stony corals, and sea anemones might have the same origin and similar biological functions. It was shown that ALPs from hydra and sea anemones (*N. vectensis*) are involved in prey disintegration after predation by lysing cells through pore formation on membranes[50,51]. The microbial communities of cold seeps are very different from those of other seafloor ecosystems[52]. Moreover, some of these microbes have unique cellular structures that might be difficult to disintegrate[53], which impedes nutrient acquisition of *C. heheva* from free-living microbes of cold seeps. Therefore, the expansion of the ALP family might have contributed to microbe digestion in *C. heheva*, which in turn facilitated its adaptation to cold seep environments.

**Positively selected genes.** To better understand the genetic basis of its adaptation to a deep-sea reducing environment, we searched for positively selected genes (PSGs) in *C. heheva*. Compared with 6 other echinoderms, 27 PSGs were identified in the *C. heheva* genome (Supplementary Table 11). Four hypoxia-related genes (pyruvate kinase M2, *PKM2*; phospholysine phosphohistidine inorganic pyrophosphate phosphatase, *LHPP*; poly(A)-specific ribonuclease subunit PAN2, *PAN2*; and ribosomal RNA processing 9, *RRP9*) were identified as PSGs in *C. heheva*[54–57]. PKM2 promotes transactivation of HIF-1 target genes by directly interacting with the HIF-1α subunit. In addition, the transcription of the *PKM2* gene is activated by HIF-1. This positive-feedback loop increases glycolysis and lactate production and decreases oxygen consumption under hypoxic conditions[54]. LHPP interacts with PKM2 to induce ubiquitin-mediated degradation of PKM2 and impede the glycolysis and respiration under hypoxia[55]. Thus, selection against these two interacting genes (*PKM2* and *LHPP*) might play a key role in the hypoxic adaptation in *C. heheva*. Interestingly, the LHPP was also subject to positive selection in cetaceans, which are hypoxia-tolerant mammals[58]. Furthermore, both *C. heheva* and cetaceans have the same amino acid substitution at position 118 of the LHPP protein (Fig. 6), which indicates a possible convergent evolution in the *LHPP* during the adaptation of cetaceans and *C. heheva* to hypoxic environments. A positively charged amino acid (histidine, H) in two shallow-water

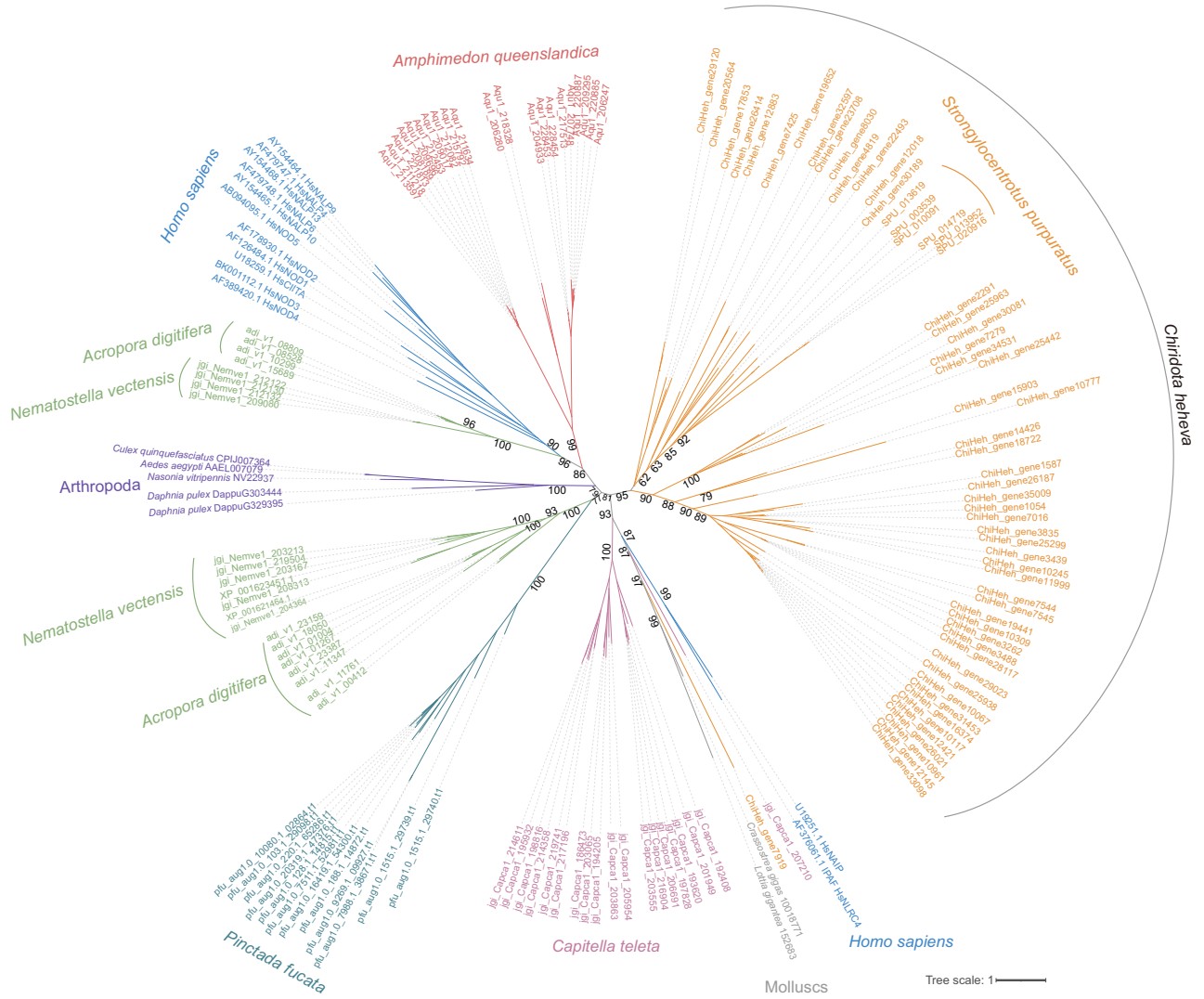

**Fig. 4 Evolutionary relationships among *C.heheva* NLRs and other representative metazoan NLRs.** The unrooted phylogenetic tree was reconstructed based on the NACHT-domain sequences using a maximum-likelihood method. The values near the nodes are ultrafast bootstrap (UFBoot) values. NLRs from different types of species are highlighted by branches of different colors. The species name is shown near the corresponding lineage.

holothurians is substituted to a negatively charged one (aspartic acid, D) in *C. heheva* at this position, which might cause a conformation change that contributes to the hypoxic adaptation in *C. heheva*. To study the potential underlying structural effects of this substitution, we predicted the three-dimensional structures of LHPP from echinoderms (Supplementary Fig. 4). The substitution, which is located in an α-helix, does not change the conformation of LHPP. The effect of the substitution of the LHPP protein needs to be further investigated.

A large number of metazoans reside in cold seeps and hydrothermal vents, which are challenging environments with high concentration of toxic compounds and chronic hypoxia[59,60]. Several physiological and molecular modifications of the respiratory system have been identified to cope with hypoxia in organisms living in these environments[60]. The concentration of hemoglobin, which is an oxygen-binding protein, is higher in seep- and vent-dwelling species than closely related species living in well-oxygenated environments[60]. In addition, hemoglobins from deep-sea organisms have higher affinity for oxygen than hemoglobins from shallow-water relatives[61]. This facilitates seep- and vent-dwelling species to thrive in the extreme environments by improving the efficiency of oxygen transportation. Four hypoxia-related genes (*PKM2*, *PAN2*,

*LHPP*, and *RRP9*) were identified to be positively selected in *C. heheva*. PKM2 increases glycolysis and decreases oxygen consumption by promoting transactivation of HIF-1 target genes through directly interacting with the HIF-1α subunit under hypoxic conditions[54]. This suggests that animals living in deep-sea chemosynthetic environments might also adapt to hypoxic conditions through reprogramming glucose metabolism. Intriguingly, *LHPP* gene was subjected to positive selection in both *C. heheva* and cetaceans. This indicates a possible convergent evolution, in which echinoderms and mammals utilize similar strategies to cope with hypoxic challenges.

## Methods

**Sample collection and genome sequencing**. The *C. heheva* sample used in this study was collected using manned submersible *Shenhaiyongshi* from the Haima cold seep in the South China Sea (16° 73.228′ N, 110° 46.143′ E, 1385 m deep) on August 2, 2019. The *C. heheva* individuals were kept in an enclosed sample chamber placed in the sample basket of the submersible. Once the samples were brought to the upper deck of the mothership, the muscle of the individuals was dissected, cut into small pieces, and immediately stored at −80 °C. The samples were then transported to Sun Yat-sen University on dry ice and stored at −80 °C until use.

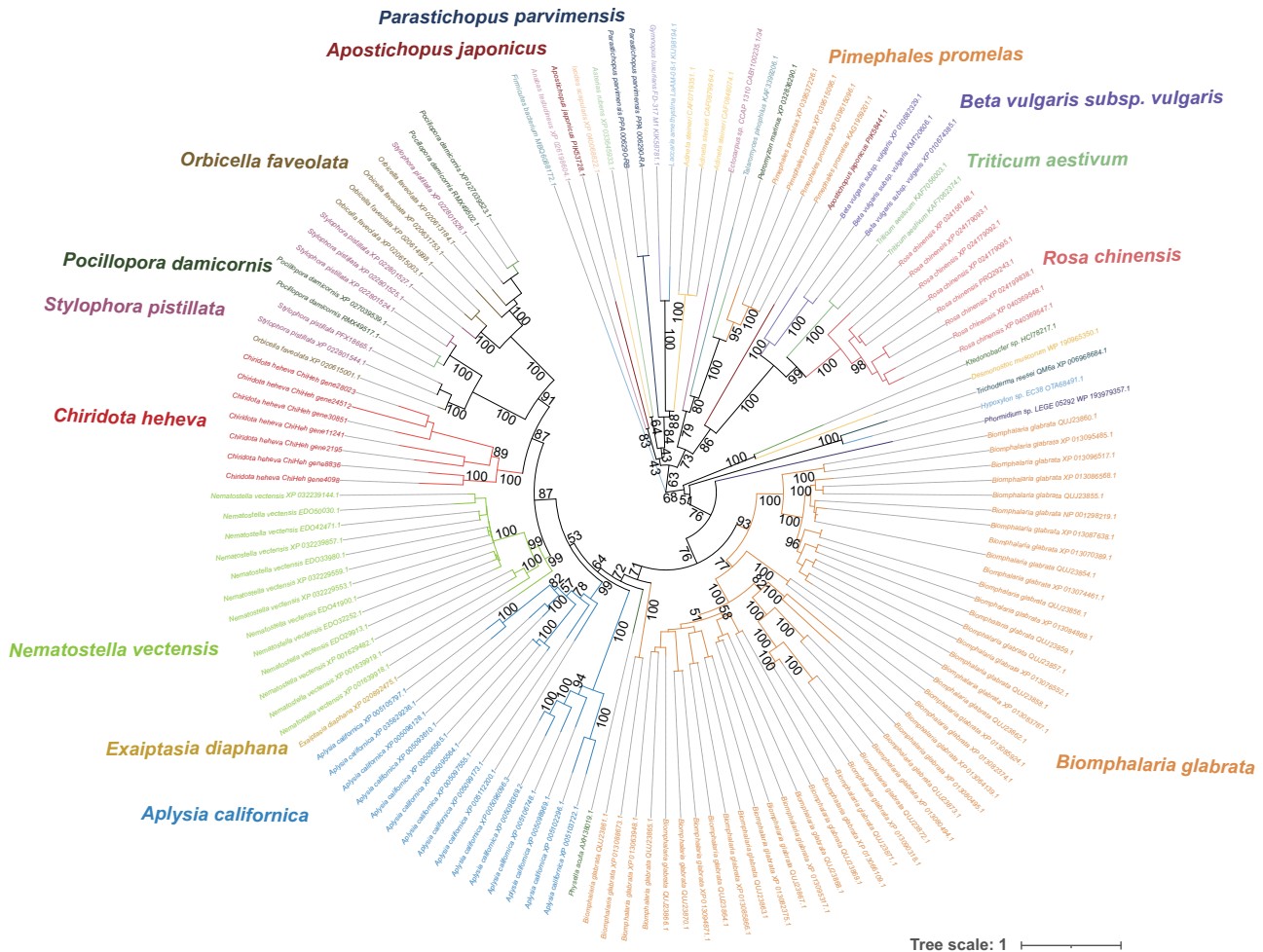

**Fig. 5 Evolutionary relationship with aerolysin-like proteins (ALPs) from *C. heheva* and other species.** The unrooted phylogenetic tree was reconstructed using a maximum-likelihood method. The values near the nodes are ultrafast bootstrap (UFBoot) values. ALPs from different types of species are highlighted by branches of different colors. The species name is shown near the corresponding lineage. ALPs from *C. heheva* do not cluster with ALPs from other echinoderms (*A. japonicus*, *P. parvimensis*), but with the ones from sea anemones (*N. vectensis*, *E. diaphana*).

To construct Nanopore sequencing library, high-molecular-weight genomic DNA was prepared by the CTAB method. The quality and quantity of the DNA were measured via standard agarose-gel electrophoresis and with a Qubit 4.0 Fluorometer (Invitrogen). Sequencing library was constructed and sequenced by Nanopore PromethION platform (Oxford Nanopore Technologies). Additionally, DNA was extracted to construct Illumina sequencing library. The quality and quantity of the DNA were measured via standard agarose-gel electrophoresis and with a Qubit 2.0 Fluorometer (Invitrogen). Sequencing library was constructed and sequenced by Illumina Novaseq platform (Illumina).

**Mitochondrial and nuclear genome assembly**. Low-quality (reads with ≥10% unidentified nucleotide and/or ≥ 50% nucleotides having phred score < 5) and sequencing-adapter-contaminated Illumina reads were filtered and trimmed with Fastp (v0.21.0)[62] to obtain high-quality Illumina reads, which were used in the following analyses. Mitochondrial genome of *C. heheva* was assembled using the two-step mode of mitoZ (v2.4)[63] with the high-quality Illumina reads. The assembled genome was annotated using mitoZ (v2.4) with parameter "–clade Echinodermata".

The size and heterozygosity of *C. heheva* genome were estimated using high-quality Illumina reads by *k*-mer frequency-distribution method. The number of *k*-mers and the peak depth of *k*-mer sizes at 17 was obtained using Jellyfish (v2.3.0)[64] with the -*C* setting. Genome size was estimated based on the *k*-mer analysis as described previously[65]. The heterozygosity of *C. heheva* genome was determined by fitting the *k*-mer distribution of *Arabidopsis thaliana* using Kmerfreq implemented in SOAPdenovo2 (r242)[66].

Low-quality Nanopore reads were filtered using custom Python script. Two draft-genome assemblies were generated using filtered Nanopore reads with Shasta (v0.4.0)[67] and WTDBG2 (v2.5)[68], respectively. The contigs of the two draft assemblies were subject to error correction using filtered Nanopore reads with Racon (v1.4.16)[69] three times. The corrected contigs were then polished with high-

quality Illumina reads with Pilon (v1.23)[70] three times. The error-corrected contigs of Shasta assembly and WTDBG2 assembly were assembled into longer sequences using quickmerge (v0.3)[71]. The merged contigs were subject to error correction using filtered Nanopore reads with Racon three times, and then using high-quality Illumina reads with Pilon three times. As the heterozygosity of *C. heheva* genome is high, haplotypic duplications in the assembled genome were identified and removed using purge_dups (v1.2.3)[72]. The completeness and quality of the assembly was evaluated using BUSCO (v4.0.5)[73] against the conserved Metazoa dataset (obd10), and SQUAT with high-quality Illumina reads[74].

**Genome annotation**. Repetitive elements in the assembly were identified by de novo predictions using RepeatMasker (v4.1.0) (https://www.repeatmasker.org/). A de novo repeat library for *C. heheva* was built using RepeatModeler (v2.0.1)[75]. To identify repetitive elements, sequences from the *C. heheva* assembly were aligned to the *de novo* repeat library using RepeatMasker (v4.1.0). Additionally, repetitive elements in *C. heheva* genome assembly were identified by homology searches against known repeat databases using RepeatMasker (v4.1.0), and then using RepeatMasker (v4.1.0). A repeat landscape of *C. heheva* genome was obtained using an R script that was modified from https://github.com/ValentinaBoP/TransposableElements. To compare the proportion and composition of repetitive elements among the genomes of echinoderms, genome sequences of *Strongylocentrotus purpuratus* (GCA_000002235.3), *Lytechinus variegatus* (GCA_000239495.2), *Acanthaster planci* (GCA_001949165.1), and *Anneissia japonica* (GCA_011630105.1) were downloaded from NCBI. Genome sequence of *Parastichopus parvimensis* was downloaded from echinobase (http://bouzouki.bio.cs.cmu.edu/Echinobase/PpDownloads, retrieved September 2021). Repetitive elements in the genomes of these species were identified by homology searches against known repeat databases using RepeatMasker (v4.1.0). The proportion and composition of repetitive elements of *Apostichopus japonicus* was obtained from Li et al. (2018)[11].

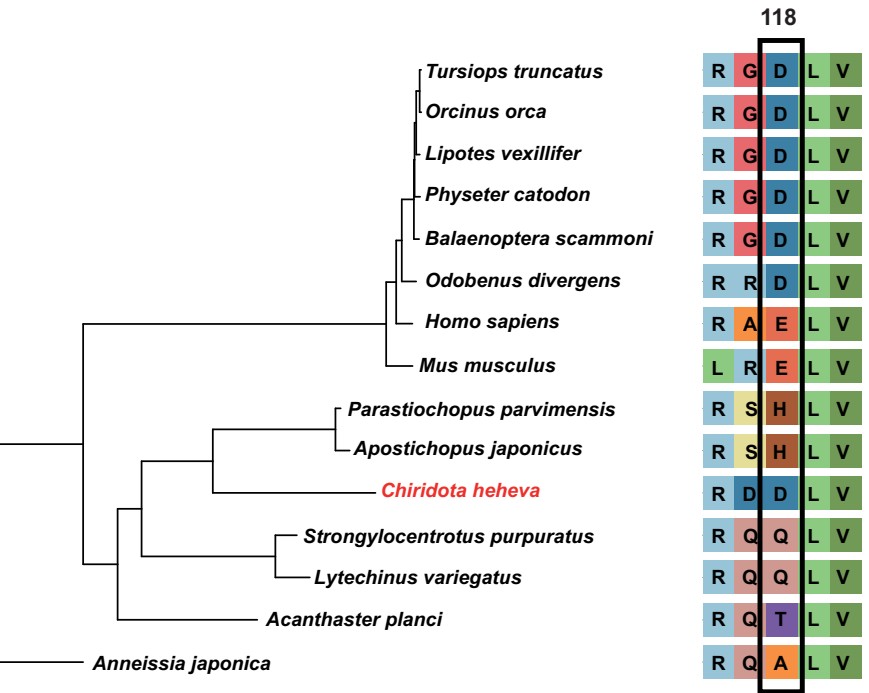

**Fig. 6 A possible amino acid substitution of LHPP that contributed to hypoxic adaptation in *C. heheva* and cetaceans.** The maximum-likelihood phylogenetic tree of cetaceans, *C. heheva*, and other echinoderms was reconstructed using 598 single-copy orthologs. *C. heheva* and cetaceans, which are tolerant to hypoxia, have the same amino acid substitution at position 118 of LHPP protein.

We applied a combination of homology-based and *de novo* predication methods to build consensus-gene models for the *C. heheva* genome assembly. For homology-based gene prediction, protein sequences of *Helobdella robusta*, *Lytechinus variegatus*, *Strongylocentrotus purpuratus*, *Dimorphilus gyrociliatus*, *Apostichopus japonicus* and *Acanthaster planci* were aligned to the *C. heheva* genome assembly using tblastn. The exon–intron structures then were determined according to the alignment results using GenomeThreader (v1.7.0)[76]. In addition, *de novo* gene prediction was performed using Augustus (v3.3.2)[77], with the parameters obtained by training the software with protein sequences of *Drosophila melanogaster* and *Parasteatoda tepidariorum*. Two sets of gene models were integrated into a nonredundant consensus-gene set using EvidenceModeler (v1.1.1)[78]. To identify functions of the predicted proteins, we aligned the *C. heheva* protein models against NCBI NR, trEMBL, and SwissProt database using blastp (E-value threshold: $10^{-5}$), and against eggNOG database[79] using eggNOG-Mapper[80]. In addition, KEGG annotation of the protein models was performed using GhostKOALA[81].

**Phylogenomic analysis.** Protein sequences of 15 metazoan species (A. planci, S. purpuratus, Lytechinus variegatus, A. japonicus, Anneissia japonica, Saccoglossus kawalevskii, Branchiostoma floridae, Ciona intestinalis, Danio rerio, Gallus gallus, H. robusta, Mus musculus, Pelodiscus sinensis, Petromyzon marinus, and Xenopus laevis) proteins were downloaded from NCBI. Protein sequences of Parastichopus parvimensis were downloaded from Echinobase[12]. OrthoMCL (v2.0.9)[82] was applied to determine and cluster gene families among these 16 metazoan species and C. heheva. Gene clusters with >100 gene copies in one or more species were removed. Single-copy othologs in each gene cluster were aligned using MAFFT (v7.310)[83]. The alignments were trimmed using ClipKit (v1.1.3)[84] with "gappy" mode. The phylogenetic tree was reconstructed with the trimmed alignments using a maximum-likelihood method implemented in IQ-TREE2 (v2.1.2)[85] with H. robusta as outgroup. The best-fit substitution model was selected by using Mod-elFinder algorithm[86]. Branch supports were assessed using the ultrafast bootstrap (UFBoot) approach with 1000 replicates[87].

To estimate the divergent time among echinoderms, single-copy orthologs were identified among *A. japonica*, *A. planci*, *A. japonicus*, *P. parvimensis*, *C. heheva*, *L. variegatus*, and *S. purpuratus* after running OrthoMCL pipeline as mentioned above. Single-copy orthologs were aligned using MAFFT (v7.310), trimmed using ClipKit (v1.1.3) with 'gappy' mode, and concatenated using PhyloSuite (v1.2.2)[88]. Divergent time among 7 echinoderms were estimated using the concatenated alignment with MCMCtree module of the PAML package (v4.9)[89]. MCMCtree analysis was performed using the maximum-likelihood tree that was reconstructed by IQ-TREE2 as a guide tree and calibrated with the divergent time obtained from TimeTree database (minimum = 193 million years and soft maximum = 350 million years between *L. variegatus* and *S. purpuratus*)[90].

**Demographic inference of *C. heheva* and *A. japonicus*.** Paired-end Illumina reads of *A. japonicus* were downloaded from NCBI SRA database[11]. The reads of *A. japonicus* were filtered and trimmed with fastp (v0.21.0). The Illumina clean reads of *C. heheva* and *A. japonicus* were aligned to the respective reference-genome assembly using BWA (v0.7.17)[91] with "mem" function. Genetic variants were identified using Samtools (v1.9)[92]. Whole-genome consensus sequence was generated with the genetic variants using Samtools (v 1.9). PSMC (v0.6.5)[93] was used to infer the demographic history of *C. heheva* and *A. japonicus* using the whole genome consensus sequences. The substitution mutations rate and generation time of *C. heheva* and *A. japonicus* was set to $1.0 \times 10^{-8}$ and 3 years according to the previous study of *A. planci*[94].

**Homeobox gene analysis.** Homeobox genes in *C. heheva* genome were identified by following the procedure described previously[95]. Homeodomain sequences, which were retrieved from HomeoDB database (http://homeodb.zoo.ox.ac.uk)[96], were aligned to *C. heheva* genome assembly using tblastn. Sequences of the candidate homeobox genes were extracted based on the alignment results. The extracted sequences were aligned against NCBI NR and HomeoDB database to classify the homeobox genes.

**Identification of NOD-like receptors (NLRs) in *C. heheva*.** We used HMMER (v3.1)[97] to search against the proteome of *C. heheva* with the HMM profile of NACHT domain (PF05729) retrieved from Pfam 34.0 as the query and an e cutoff value of 0.01. Proteins identified by the HMM search were retrieved from the proteome and aligned with 964 representative proteins from eukaryotes and prokaryotes[98], and other representative metazoan NLRs[43] using hmmalign method implemented in HMMER (v3.1) based on the STAND NTPase domain. The alignment was refined by manual editing. The large-scale phylogenetic analysis was performed using an approximate maximum likelihood method implemented in FastTree[99]. Representative SWACOS and MalT NTPases were used as outgroups[98]. Significant hit clustering with metazoan NLRs was regarded as NLRs, and protein-domain organizations were annotated through hmmscan method implemented in HMMER (v3.1).

**Phylogenetic analysis of Chiridota NLRs and other representative metazoan NLRs.** To explore the evolutionary relationships among *C. heheva* NLRs and other representative metazoan NLRs, we reconstructed the phylogenetic tree of NLRs. The NACHT domains of *C. heheva* NLRs and representative metazoan NLRs were aligned using MAFFT (v7.310), and then refined by manual editing. The representative metazoan NLRs were chosen from literature[43]. The phylogenetic tree was reconstructed using a maximum-likelihood method implemented in IQ-TREE2 (v2.1.2). The best-fit substitution model was selected by using ModelFinder

algorithm. Branch supports were assessed using the UFBoot approach with 1000 replicates.

**Gene-family expansion and contraction analysis**. r8s (v1.7)[100] was applied to obtain the ultrametric tree of 7 echinoderm species, which is calibrated with the divergent time between *A. planci* and *S. purpuratus* (541 mya) obtained from TimeTree database. CAFÉ (v5)[101] was applied to determine the significance of gene-family expansion and contraction among 7 echinoderm species based on the ultrametric tree and the gene clusters determined by OrthoMCL (v2.0.9). The divergence time reported by TimeTree database might not be precise as it is a consensus of divergence times estimated in previous studies. Therefore, we repeated the analysis twice, in which the divergence time between *A. planci* and *S. purpuratus* was set to 461 mya and 495 mya according to the previous studies, respectively[102,103]. All the three analyses had the same result.

We used HMMER (v3.3.2) to search against NCBI nonredundant protein database (accessed on July 2021) with the HMM profile of aerolysin domain (PF01117) retrieved from Pfam 34.0 as the query and a *e* cutoff value of 0.001. Proteins identified by the HMM search were retrieved and filtered for the ones that have less than 75 residues. The filtered proteins were aligned with aerolysin-like proteins (ALPs) from *C. heheva*, *A. japonicas*, and *P. parvimensis* using MAFFT (v7.310) and trimmed using ClipKit (v1.1.3) with 'gappy' mode. The phylogenetic tree was reconstructed using a maximum-likelihood method implemented in IQ-TREE2 (v2.1.2). The best-fit substitution model was selected by using ModelFinder algorithm. Branch supports were assessed using the UFBoot approach with 1000 replicates.

**Identification and analysis of positively selected genes**. Branch-site models implemented in the codeml module of the PAML package is widely used to identify positively selected genes (PSGs). Thus, we identified PSGs in the *C. heheva* genome within the single-copy orthologs among 7 echinoderm species, based on the branch-site models using GWideCodeML (v1.1)[104]. *C. heheva* was set as the 'foreground' phylogeny, and the other species were set as the 'background' phylogeny. An alternative branch site model (Model = 2, NSsites = 2, and fix_-omega = 0) and a neutral branch site model (Model = 2, NSsites = 2, fix_omega = 1, and omega = 1) were tested. Genes with Bayesian empirical Bayes (BEB) sites > 90% and a corrected *P*-value < 0.1 were identified to have been subject to positive selection.

To investigate LHPP gene evolution, sequences of LHPP from 8 mammals (*Odobenus rosmarus*, *Orcinus orca*, *Lipotes vexillifer*, *Tursiops truncates*, *Physeter catodon*, *Balaenoptera acutorostrata*, *Mus musculus*, and *Homo sapiens*) and 7 echinoderms (*A. japonica*, *A. planci*, *A. japonicus*, *P. parvimensis*, *C. heheva*, *L. variegatus*, and *S. purpuratus*) were aligned using MAFFT (v7.310). To reconstruct the phylogenetic tree, OrthoMCL (v2.0.9)[82] was applied to determine and cluster gene families among these 15 species. Gene clusters with >100 gene copies in one or more species were removed. Single-copy othologs in each gene cluster were aligned using MAFFT (v7.310)[83]. The alignments were trimmed using ClipKit (v1.1.3)[84] with "gappy" mode. The phylogenetic tree was reconstructed with the trimmed alignments using a maximum-likelihood method implemented in IQ-TREE2 (v2.1.2)[85]. *H. robusta* was used as outgroup. The best-fit substitution model was selected by using ModelFinder algorithm[86]. The three-dimensional structure of a protein provides important information for understanding its biochemical function and interaction properties in molecular detail. In this study, the three-dimensional structure of four LHPP proteins from (*O. orca*, *H. sapiens*, *A. japonicus* and *C. heheva*) was generated through homology modeling using the SWISS-MODEL workspace (http://swissmodel.expasy.org/workspace/)[105].

**Statistics and reproducibility**. Alpha levels of 0.05 were regarded as statistically significant throughout the study, unless otherwise specified.

## Data availability
Raw reads and genome assembly are accessible in NCBI under BioProject number PRJNA752986. Assembled genome sequences are accessible under Whole Genome Shotgun project number JAIGNY000000000. Raw reads and genome assembly are also available at the CNGB Sequence Archive (CNSA) of China National GeneBank DataBase (CNGBdb) with accession number CNP0002134. The genome assembly, related annotation files, and source files for generating figures can be accessed through Figshare[106] at https://doi.org/10.6084/m9.figshare.15302229.

## Code availability
Custom script used in this study is available at Figshare[106] (https://doi.org/10.6084/m9.figshare.15302229). Versions and parameters for other software packages used in this study are described in the reporting summary and elsewhere in the "Methods."

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

## Acknowledgements

We thank Dr. Kang Ding and Dr. Zhimin Jian for leading the expedition of TS12-02, the crew of research vessel (R/V) *Tansuoyihao*, the pilot team of the manned submersible *Shenhaiyongshi*, and the onboard diving scientists for their technical support during the cruise. We gratefully acknowledge the National Supercomputing Center in Guangzhou for provision of computational resources. This study was supported by National Natural Science Foundation of China (No. 31900309), GuangDong Basic and Applied Basic Research Foundation (No. 2019A1515011644), Innovation Group Project of Southern Marine Science and Engineering Guangdong Laboratory (Zhuhai) (No. 311021006), and National Innovation and Entrepreneurship Training Project for College Student of China (No. 20201126). The funders had no role in study design, data collection and analysis, decision to publish, or preparation of the paper.

## Author contributions

M.W and J.G.H. conceived of the project and designed research; J.H. collected the sample; P.T., L.Z, Y.M., G.T., Q.C., and Q.Z. assembled and annotated the genome; L.Z., J.H., Z.G., M.W., S.Q., and H.-Y.Z. performed the evolutionary analyses; M.W., G.H., and J.G.H. wrote the paper with contribution from all authors.

## Competing interests

The authors declare no competing interests.
