## [Peer Review File · Communications Biology]

Reviewers' comments:

Reviewer #1 (Remarks to the Author):

The taxon sampling is inadequate to assess any of the phylogenetic, time tree, population genetic, or horizontal transfer conclusions.

The analyses of hox gene presence and absence is misconstrued. A single character that may be labile across echinoderms is possible and does not in itself render taxa paraphyletic or monophyletic.

Why are Sea urchin NLRs nested within the clade of Chirodota heveva? Why are Homo sapiens NLR closely related to cnidaria NLR? This relates to the notion of horizontal transfer but could also be caused by inadequate sampling and analytical techniques. The latter is a much less extraordinary claim.

Reviewer #2 (Remarks to the Author):

This article reports on the determination of the complete genome of the holothurid Chirodota heveva and its adaptation to the extreme environment of deep-sea cold seeps, which are characterized by high hydrostatic pressure, low temperature and anoxia. Whereas other organisms that have been studied from this environment are dependent from chemosynthetic symbionts, this species is the first of a non-symbiotic detritus and suspension-feeding organism under study. Albeit holothurian genomes are still sparse, one shallow water species, the Japanese holothurian *Apostichopus japonicus* and other echinoderm genomes were available for comparative analyses and putative adaptations of *C. heveva* to the deep-sea environment. The manuscript is straightforward, methodologically sound and presents several outstanding results on the architecture of the *C. heveva* genome and possible adaptations to its extreme environment, including the evolutionary origin and expansion of the aerolysin-like protein family and the positive selection on several hypoxia-related genes. Among those, the authors detected a convergent substitution in the LHPP protein in *C. heveva* and hypoxia-tolerant Cetaceans.

I have several, mostly minor, comments and suggestions that will improve the manuscript:
1 lines 35-36: I would add some examples here.

2 lines 41-42: I suggest to list the classes according to the correct systematics including their grouping into subphyla. You are using those later in the context of your phylogenetic trees and the Hox-genes (line 192): e.g.: comprising five extant classes, Holothuroidea (feather star, subphylum Pelmatozoa), Asterozoa (starfish and brittle star, subphylum Asterozoa), Echinozoa (sea urchin and sea cucumber, subphylum Echinozoa).

3 lines 60, 130-135, 152, 154: this is confusing to the reader. Is there just one genome (*Apostichopus japonicus*) or are there two genomes (*A. japonicus* and *Parastichopus parvimensis*). Please clarify and rewrite accordingly throughout the text. Please spell out the genus name *Parastichopus* for the first time in the text. You cite the source of the *P. parvimensis* genomic data only lately in the methods (ref. #80, line 359).

4 lines 91-92: please give reference also here

5 line 141- 142: not true, the number of exons is higher in *A. japonicus* than in *C. heveva* in supplementary figure 8

6 line 152: "...supports the view that Apodida is the sister taxon to the remaining holothuriids". This conclusion is too far reaching: You only have representatives of the orders Synallactida and Apodida in your tree. The others are missing (*Elasipodida*, *Holothuriida*, *Persiculida*, *Molpadida*, *Dendrochirodida*).

7 line 154: the divergence times are markedly different between the text and figure 3a. Please check and harmonize.

8 line 167: "...when climate transition improved adaptations in echinoderms". The climate or its change do not improve anything, it sets or changes the frame of selection pressure that may result in adaptation.

9 line 214: It is a bit strange here to write "purple sea urchin as a member of the phylum Echinodermata". It was mentioned several times before as *Strongylocentrotus purpuratus* and member of Echinoidea, Echinozoa.

10 line 252: "C. heveva ALPs form a clade with ALPs from sea anemones." The tree in figure 6 appears a bit bush-like, and the bootstrap-values are by no means readable. However, it seems to me that C. heveva groups with colony-forming Scleractinia and not with solitary sea anemones (all Hexacorallia).

11 lines 276 and following: to me this is an outstanding result. I would suggest to move suppl. figure 3 to the main text. Conversely, the supplementary figure 4 is quite complex and lengthy, and I asked myself, additional information is provided by this figure. More important, I would like to encourage the authors to write more on this amino-acid substitution at position 118 in C. heveva and whales and its possible implication to hypoxia adaptation. Under physiological conditions histidin (present in the other holothurians at this position) is positively charged, while aspartic acid (C. heveva and whales) is negatively charged. This may result in a conformation change. In which region of the protein is this mutation located?

12 lines 280-281: I can understand what the authors want to express (maybe silencing of LHPP induced degradation of PMK2 and inhibition of glycolysis under hypoxia). But it is confusing to write selection against hypoxia-related genes, and to name this positive selection. Furthermore, the text ends quite abruptly here. Some conclusive sentences on the importance of the results might be appropriate.

13 Figure 1: Please give credit to the origin of the figure

14 Supplementary figure 1: Please explain the dashed blue line. *A. thaliana* is not mentioned neither in the main text nor in the suppl information.

15 Supplementary table 7: Please give accession numbers here or in methods

Reviewer #3 (Remarks to the Author):

This manuscript by Zhang and He et al searches for a genetic basis of nutrient acquisition among non-symbionts in cold seeps. To this end, they sequence a holothuroid genome (*Chiridota heveva*), and interrogate it using a suite of phylogenomic methods.

The new genome is of good quality, with 89.6% BUSCO completeness. The contig N50s exceed those of existing holothuroid genomes, and are not far off of those reported for traditional model echinoderms (e.g. *P. miniata* contig N50 = ~2.1Mb).

In general, I think the findings are of interest and within the scope of the journal, e.g. the implications of ancient global changes as drivers of demographic changes, the contributions of ALP genes towards adapting to a cold seep environment, and the identification of a potentially adaptive gene suite that has been subjected to strong selection pressures.

There was much to enjoy in this manuscript. However, there are some points which I would like to see addressed, primarily concerning the methods.

General comments:

1. Line 200. Remove the comma preceding citation 39.
2. Line 352: Please change both instances of "eggNOR" to "eggNOG".
3. Line 358: Add a space in "laevisproteins".
4. Line 359: Per the Echinobase website, a more appropriate citation to use might be doi: 10.1093/nar/gkab1005
5. Line 377: The papers that reported the divergence times can be cited (i.e. Hedges et al. 2009 for the minimum bound, and Aris-Brosou et al. 2013 for the maximum), and / or a citation for TimeTree could be used, per <http://www.timetree.org/faqs#q7>
6. The resolution for figures 5 and 6 could be improved - I found it difficult to read them.
7. Custom scripts used throughout the methods section. Will these be made publicly available?

Methodological concerns:

8. Line 319 and 324: Why run Racon and Pilon three times?
9. Line 401: Can you please address why 541Mya was used as the divergence time? The following paper shows the divergence of echinoids and asteroids to be +/- 400Mya <https://doi.org/10.1186/s12915-018-0499-2>. Your Figure 3a puts the date at 496.6Mya. The paper discusses the emergence of various echinoderm classes, and mentions that the oldest echinoderms probably emerged 530Mya <https://doi.org/10.1016/j.ydbio.2017.02.003>.
10. Line 707: The description should clarify that the *C. heheva* and *A. japonicus* genomes are being compared because they are both holothurids.

Reviewers' comments:

Reviewer #1 (Remarks to the Author):

The taxon sampling is inadequate to assess any of the phylogenetic, time tree, population genetic, or horizontal transfer conclusions.

Response: Thanks for your great comment. We tried to have a comprehensive taxon sampling for each of our analyses.

To improve the accuracy of phylogenetic reconstruction, we constructed the phylogenetic tree (Fig. 2a) and time tree (Fig. 3a) using all the single-copy orthologs among genomes of studied species. All the echinoderm species that have reference sequences were used in these two evolutionary analyses to obtain a complete set of single-copy orthologs. We agree with Reviewer #2 that it is inadequate to conclude Apodida is the sister taxon to the remaining holothuroids according to our result of phylogenetic reconstruction (Fig. 2a). We have deleted this description in the revised manuscript.

To investigate the evolution of NLR in *C. heheva* (Fig. 4 in the revised manuscript), we chose representative animals, and human (*Homo sapiens*) serves as the representative of vertebrate. In the manuscript, NLR sequences from other metazoans used for phylogenetic analysis were retrieved from a previous study (doi:10.1093/molbev/mst174), which reveals the origin of the two major NLR lineages at the last common eumetazoan ancestor.

To identify the origin of aerolysin-like proteins (ALPs) in *C. heheva*, we used HMMER to search against NCBI non-redundant protein database with the HMM profile of aerolysin domain retrieved from Pfam. We think that a set of representative aerolysin/aerolysin-like proteins can be retrieved to identify the possible origin of ALPs in *C. heheva*.

The analyses of hox gene presence and absence is misconstrued. A single character that may be labile across echinoderms is possible and does not in itself render taxa paraphyletic or monophyletic.

Response: Thank you for your excellent advice. We have deleted this section in the revised manuscript. We also modified and moved **Figure 4** to Supplementary Information (**Supplementary Figure 3**) in the revised manuscript.

Why are Sea urchin NLRs nested within the clade of Chirodota heheva? Why are *Homo sapiens* NLR closely related to cnidaria NLR? This relates to the notion of

horizontal transfer but could also be caused by inadequate sampling and analytical techniques. The latter is a much less extraordinary claim.

Response: Thanks for the great comment! The main purpose here was to analyze the NLR repertoire and evolution in *Chirodoate heheva*. We chose representative animals, and human (*Homo sapiens*) serves as the representative of vertebrate. In the manuscript, NLR sequences from other metazoans used for phylogenetic analysis were retrieved from a previous study (doi:10.1093/molbev/mst174), which reveals the origin of the two major NLR lineages at the last common eumetazoan ancestor. The sea urchin NLRs nested within the clade of *C. heheva* is largely due to frequent lineage-specific NLR gains and losses.

Reviewer #2 (Remarks to the Author):

This article reports on the determination of the complete genome of the holothroid *Chridota heveva* and its adaptation to the extreme environment of deep-sea cold seeps, which are characterized by high hydrostatic pressure, low temperature and anoxia. Whereas other organisms that have been studied from this environment are dependent from chemosynthetic symbionts, this species is the first of a non-symbiotic detritus and suspension-feeding organism under study. Albeit holothurian genomes are still sparse, one shallow water species, the Japanese holothurian *Apostichus japonicus* and other echinoderm genomes were available for comparative analyses and putative adaptations of *C. heveva* to the dep-sea environment. The manuscript is straightforward, methodologically sound and presents several outstanding results on the architecture of the *C. heveva* genome and possible adaptations to its extreme environment, including the evolutionary origin and expansion of the aerolysin-like protein family and the positive selection on several hypoxia-related genes. Among those, the authors detected a convergent substitution in the LHPP protein in *C. heveva* and hypoxia-tolerant Cetaceans.

I have several, mostly minor, comments and suggestions that sill may improve the manuscript:

1 lines 35-36: I would add some examples here.

Response: Thank you for your suggestion. We have listed gene names at Line 35 of the revised manuscript:

“Moreover, several hypoxia-related genes (*PKM*, *LHPP*, *PAN2* and *RRP9*) were subject to positive selection in the genome of *C. heheva*, which contributes to their adaptation to hypoxic environments.”

2 lines 41-42: I suggest to list the classes according to the correct systematics including their grouping into subphyla. You are using those later in the context of your phylogenetic trees and the Hox-genes (line 192): e.g.: comprising five extant classes, Holothuroidea (feather star, subphylum *Pelmatozoa*), Asterozoa and Ophiurozoa (starfish and brittle star, subphylum *Asterozoa*), Echinozoa and Holothurozoa (sea urchin and sea cucumber, subphylum *Echinozoa*).

Response: Thanks for your excellent advice. We have revised the manuscript according to your suggestion from Line 40 to Line 43:

“Echinodermata is a phylum of marine animals comprising 5 extant classes, including Holothuroidea (feather star, subphylum Pelmatozoa), Asterozoa (starfish and brittle star, subphylum Asterozoa), Echinozoa (sea urchin and sea cucumber, subphylum Echinozoa).”

3 lines 60, 130-135, 152, 154: this is confusing to the reader. Is there just one genome (*Apostichopus japonicus*) or are there two genomes (*A. japonicus* and *Parastichopus parvimensis*). Please clarify and rewrite accordingly throughout the text. Please spell out the genus name *Parastichopus* for the first time in the text. You cite the source of the *P. parvimensis* genomic data only lately in the methods (ref. #80, line 359).

Response: Thank you for your suggestion. The genomes of two shallow-water holothurians (*Apostichopus japonicus* and *Parastichopus parvimensis*) have been assembled and analyzed. We have added a description in the revised manuscript at Line 61:

“As the genomes of only two shallow-water holothurians (*Apostichopus japonicus* and *Parastichopus parvimensis*) have been assembled and analyzed”

We have also cited the source of *P. parvimensis* genome (doi: 10.1093/nar/gkab1005) at Line 62 of the revised manuscript.

4 lines 91-92: please give reference also here

Response: Thank you for your suggestion. The result was derived from our analysis. We have modified the description in the revised manuscript at Line 93:

“Evolutionary analyses revealed that the ancestor of *C. heheva* diverged from the ancestors of two shallow-water holothurians (*A. japonicus* and *P. parvimensis*) approximately 429 Ma ago.”

5 line 141- 142: not true, the number of exons is higher in *A. japonicus* than in *C. heheva* in supplementary figure 8

Response: Thank you for your advice. We have deleted the description of exon number in the revised manuscript at Line 142:

“With longer exon and intron, genes in *C. heheva* are longer than the ones in *A. japonicus*”

6 line 152: “...supports the view that Apodida is the sister taxon to the remaining holothuriids”. This conclusion is too far reaching: You only have representatives of

the orders Synallactida and Apodida in your tree. The others are missing (Elasipodida, Holothuriida, Persiculida, Molpadida, Dendrochirotida).

Response: Thank you for your suggestion. We have deleted the description in the revised manuscript at Line 151:

“*Chiridota heheva* appeared sister to two other holothurians.”

7 line 154: the divergence times are markedly different between the text and figure 3a. Please check and harmonize.

Response: Thank you for your comment. We have checked and corrected the mistake in the revised manuscript at Line 154:

“The divergence time of *A. japonica* and other echinoderms was estimated to be approximately 539 million years (Ma)”

8 line 167: “...when climate transition improved adaptations in echinoderms”. The climate or its change do not improve anything, it sets or changes the frame of selection pressure that may result in adaptation.

Response: Thank you for your great suggestion. We have deleted this statement in the revised manuscript at Line 167.

9 line 214: It is a bit strange here to write “purple sea urchin as a member of the phylum Echinodermata”. It was mentioned several times before as *Strongylocentrotus purpuratus* and member of Echinoidea, Echinozoa.

Response: Thanks! We have deleted this statement in the revised manuscript at Line 199.

10 line 252: “*C. heveva* ALPs form a clade with ALPs from sea anemones.” The tree in figure 6 appears a bit bush-like, and the bootstrap-values are by no means readable. However, it seems to me that *C. heveva* groups with colony-forming Scleractinia and not with solitary sea anemones (all Hexacorallia).

Response: Thank you for your suggestion. We have modified **Figure 6 (Figure 5 in the revised manuscript)** to make it easier to read.

The original description at Line 252 is not precise. We have changed the description in the revised manuscript at Line 238:

“*Chiridota heheva* ALPs form a clade with ALPs from stony corals (*Stylophora pistillata*, *Pocillopora damicornis*, and *Orbicella faveolata*) and sea anemones (*Nematostella vectensis* and *Ecaiptasia diaphana*). This indicates that ALPs from *C. heheva*, stony corals, and sea anemones might have same origin and similar biological functions.”

11 lines 276 and following: to me this is an outstanding result. I would suggest to move suppl. figure 3 to the main text. Conversely, the supplementary figure 4 is quite complex and lengthy, and I asked myself, additional information is provided by this figure. More important, I would like to encourage the authors to write more on this amino-acid substitution at position 118 in *C. heveva* and whales and its possible implication to hypoxia adaptation. Under physiological conditions histidin (present in the other holothurians at this position) is positively charged, while aspartic acid (*C. heveva* and whales) is negatively charged. This may result in a conformation change. In which region of the protein is this mutation located?

Response: Thank you for your excellent advice. We moved Supplementary Figure 3 to the main text (Figure 6 in the revised manuscript). Additionally, Supplementary Figure 4 has been removed in the revised manuscript.

We predicted the three-dimensional structures of LHPP from echinoderms according to your suggestion (Supplementary Figure 4 in the revised manuscript). The substitution, which is located in an α helix, does not change the conformation of LHPP. We have added a description in the revised manuscript from Line 264 to Line 271.

12 lines 280-281: I can understand what the authors want to express (maybe silencing of LHPP induced degradation of PMK2 and inhibition of glycolysis under hypoxia). But it is confusing to write selection against hypoxia-related genes, and to name this positive selection.

Furthermore, the text ends quite abruptly here. Some conclusive sentences on the importance of the results might be appropriate.

Response: Thank you for your suggestion. We did not describe our results clearly in the manuscript. In total, 27 positively selected genes (PSGs) were identified in *C. heheva* genome. The four hypoxia-related genes (*PKM2*, *PAN2*, *LHPP*, *RRP9*) are among these 27 PSGs. Thus, we named these genes positively selected genes. We have added a description in the revised manuscript from Line 253 to Line 264. We also discussed the result in the revised manuscript from Line 273 to Line 289.

13 Figure 1: Please give credit to the origin of the figure

Response: Thank you for your suggestion. We have added a sentence in the revised manuscript at Line 760 to indicate that the photo (**Fig. 1b**) was taken by one of the authors, Dr. Jian He.

14 Supplementary figure1: Please explain the dashed blue line. *A. thaliana* is not mentioned neither in the main text nor in the suppl information.

Response: Thank you for your advice. We have added a sentence in the legend of Supplementary Figure 1:

“The heterozygosity of *C. hevea* genome was estimated using Kmerfreq by fitting the distribution of *Arabidopsis thaliana*.”

We have also added a description in method section of the revised manuscript at Line 321:

“The heterozygosity of *C. hevea* genome was determined by fitting the *k*-mer distribution of *Arabidopsis thaliana* using Kmerfreq implemented in SOAPdenovo2 (r242).”

15 Supplementary table 7: Please give accession numbers here or in methods.

Response: Thank you for your suggestion. We have cited the references in Supplementary table 7 and provided accession numbers in the method section from Line 347 to 357 of the revised manuscript.

Reviewer #3 (Remarks to the Author):

This manuscript by Zhang and He et al searches for a genetic basis of nutrient acquisition among non-symbionts in cold seeps. To this end, they sequence a holothuroid genome (*Chiridota heheva*), and interrogate it using a suite of phylogenomic methods.

The new genome is of good quality, with 89.6% BUSCO completeness. The contig N50s exceed those of existing holothuroid genomes, and are not far off of those reported for traditional model echinoderms (e.g. *P. miniata* contig N50 = ~2.1Mb).

In general, I think the findings are of interest and within the scope of the journal, e.g. the implications of ancient global changes as drivers of demographic changes, the contributions of ALP genes towards adapting to a cold seep environment, and the identification of a potentially adaptive gene suite that has been subjected to strong selection pressures.

There was much to enjoy in this manuscript. However, there are some points which I would like to see addressed, primarily concerning the methods.

General comments:

1. Line 200. Remove the comma preceding citation 39.

Response: Thank you for your suggestion. We removed the section according to Reviewer #1's suggestion.

2. Line 352: Please change both instances of "eggNOR" to "eggNOG".

Response: Thank you for pointing out our mistake. We have changed the words in the revised manuscript at Line 370.

3. Line 358: Add a space in "laevisproteins".

Response: Thank you for pointing out our mistake. We have revised in the manuscript at Line 377.

4. Line 359: Per the Echinobase website, a more appropriate citation to use might be doi: 10.1093/nar/gkab1005

Response: Thank you for your suggestion. We have cited the paper in the revised manuscript at Line 378.

5. Line 377: The papers that reported the divergence times can be cited (i.e. Hedges et al. 2009 for the minimum bound, and Aris-Brosou et al. 2013 for the maximum), and / or a citation for TimeTree could be used, per <http://www.timetree.org/faqs#q7>

Response: Thank you for your advice. We have cited TimeTree (DOI: 10.1093/molbev/msx116) in the revised manuscript at Line 397.

6. The resolution for figures 5 and 6 could be improved - I found it difficult to read them.

Response: Thank you for your suggestion. We have redrawn **Figure 5** (Figure 4 in the revised manuscript) and **Figure 6** (Figure 5 in the revised manuscript) and make them easier to read and interpret.

7. Custom scripts used throughout the methods section. Will these be made publicly available?

Response: Thank you for your comment. We have uploaded the script to Figshare at <https://doi.org/10.6084/m9.figshare.15302229>. We also added a description in the Data availability section at Line 744 of the revised manuscript:

“The genome assembly, related annotation files, and custom script are available at Figshare (<https://doi.org/10.6084/m9.figshare.15302229>).”

Methodological concerns:

8. Line 319 and 324: Why run Racon and Pilon three times?

Response: Thank you for your comment. As the error rate of Nanopore sequencing technology is relatively high, it is a common practice to correct the sequencing errors and improve the quality of assembled genome using Racon and Pilon. We found that running Racon and Pilon three times can further improve the result of error correction. In addition, this procedure has also been used in the assembly of several high-quality

reference sequences (eg. doi: 10.1101/gr.266429.120v2, doi: 10.1093/molbev/msaa246, doi: 10.1038/s41559-020-1166-x, doi: 10.1038/s41467-018-07271-1, doi: 10.1038/s41467-020-15522-3).

9. Line 401: Can you please address why 541Mya was used as the divergence time? The following paper shows the divergence of echinoids and asteroids to be +/- 400Mya <https://doi.org/10.1186/s12915-018-0499-2>. Your Figure 3a puts the date at 496.6Mya. The paper discusses the emergence of various echinoderm classes, and mentions that the oldest echinoderms probably emerged 530Mya <https://doi.org/10.1016/j.ydbio.2017.02.003>.

Response: The divergence time was obtained from the TimeTree database. We have added a description in the revised manuscript at Line 443:

“which is calibrated with the divergent time between *A. planci* and *S. purpuratus* (541 mya) obtain from TimeTree database.”

10. Line 707: The description should clarify that the *C. heheva* and *A. japonicus* genomes are being compared because they are both holothurids.

Response: Thank you for your suggestion. We have added a description in the revised manuscript at Line 752:

“Table 1 Genome assembly statistics of deep-sea holothurian (*C. heheva*) and shallow-water holothurian (*A. japonicus*)”

REVIEWERS' COMMENTS:

Reviewer #2 (Remarks to the Author):

The authors carefully revised their manuscript and addressed my comments and suggestions. I am pleased that they followed my suggestion to investigate a possible conformational change in the LHPP protein due to the exchange of histidine to aspartic acid.

The authors added some sentences on the importance on their findings at the end of the manuscript. Although these are a bit redundant to the paragraph before, they still put the results in a broader evolutionary context.

Some minor comments:

1 lines 35-26: The authors added the appropriate abbreviations of the hypoxia-related genes.

However, it might be helpful to the reader to write out the names here.

2 line 142: with longer exons and introns

3 line 251: replace "identified" by "searched for"

4 line 257: induce

5 line 267: contributes

Reviewer #3 (Remarks to the Author):

My comments have largely been addressed satisfactorily. Per my point 9 on the divergence time, I would like to note that TimeTree uses a consensus of previous divergence time estimates for its reporting. In looking at the times reported for the Echinoidea / Asteroidea split, the last study to report a time of >500Mya was published in 2005. All of the rest reports estimates of <500Mya; perhaps this is due to the availability of more computational resources and sequencing technologies? It seems like these older studies are skewing the TimeTree estimate and resulting in a more ancient estimated value. I would be satisfied if a short note was added to this analysis which outlines such limitations of using a consensus estimate generated from a range of older and newer publications.

Reviewers' comments:

Reviewer #2 (Remarks to the Author):

The authors carefully revised their manuscript and addressed my comments and suggestions. I am pleased that they followed my suggestion to investigate a possible conformational change in the LHPP protein due to the exchange of histidine to aspartic acid.

The authors added some sentences on the importance on their findings at the end of the manuscript. Although these are a bit redundant to the paragraph before, they still put the results in a broader evolutionary context.

Some minor comments:

1 lines 35-26: The authors added the appropriate abbreviations of the hypoxia-related genes. However, it might be helpful to the reader to write out the names here.

Response: Thank you for your suggestion. We have added the gene names in the abstract.

2 line 142: with longer exons and introns

Response: Thank you. We have corrected the mistake.

3 line 251: replace “identified” by “searched for”

Response: Thank you. We have replaced the term in the manuscript.

4 line 257: induce

Response: Thank you. We have corrected the mistake.

5 line 267: contributes

Response: Thank you. We have corrected the mistake.

Reviewer #3 (Remarks to the Author):

My comments have largely been addressed satisfactorily. Per my point 9 on the divergence time, I would like to note that TimeTree uses a consensus of previous divergence time estimates for its reporting. In looking at the times reported for the Echinoidea / Asteroidea split, the last study to report a time of >500Mya was published in 2005. All of the rest reports estimates of <500Mya; perhaps this is due to the availability of more computational resources and sequencing technologies? It seems like these older studies are skewing the TimeTree estimate and resulting in a more ancient estimated value. I would be satisfied if a short note was added to this analysis which outlines such limitations of using a consensus estimate generated from a range of older and newer publications.

Response: Thank you for your excellent advice. We agree that using a consensus estimate generated from a range of older and newer publications might not be precise. Thus, we performed the CAFÉ analysis twice and set the divergence time between *A. planci* and *S. purpuratus* to 461 mya and 495 mya according to the previous studies, respectively. The results of these two analyses are the same as the one in which the divergence time was set to 541 mya. We added a short note in the revised manuscript according to your suggestion:

“The divergence time reported by TimeTree database might not be precise as it is a consensus of divergence times estimated in previous studies. Therefore, we repeated the analysis twice, in which the divergence time between *A. planci* and *S. purpuratus* was set to 461 mya and 495 mya according to the previous studies, respectively. All the three analyses had the same result.”